# Random Tessellation Forests

**Shufei Ge**[1]
shufei_ge@sfu.ca

**Shijia Wang**[2,1]
shijia_wang@sfu.ca

**Yee Whye Teh**[3]
y.w.teh@stats.ox.ac.uk

**Liangliang Wang**[1]
liangliang_wang@sfu.ca

**Lloyd T. Elliott**[1]
lloyd_elliott@sfu.ca

[1]Department of Statistics and Actuarial Science, Simon Fraser University, Canada
[2]School of Statistics and Data Science, LPMC & KLMDASR, Nankai University, China
[3]Department of Statistics, University of Oxford, UK

## Abstract

Space partitioning methods such as random forests and the Mondrian process are powerful machine learning methods for multi-dimensional and relational data, and are based on recursively cutting a domain. The flexibility of these methods is often limited by the requirement that the cuts be axis aligned. The Ostomachion process and the self-consistent binary space partitioning-tree process were recently introduced as generalizations of the Mondrian process for space partitioning with non-axis aligned cuts in the two dimensional plane. Motivated by the need for a multi-dimensional partitioning tree with non-axis aligned cuts, we propose the Random Tessellation Process (RTP), a framework that includes the Mondrian process and the binary space partitioning-tree process as special cases. We derive a sequential Monte Carlo algorithm for inference, and provide random forest methods. Our process is self-consistent and can relax axis-aligned constraints, allowing complex inter-dimensional dependence to be captured. We present a simulation study, and analyse gene expression data of brain tissue, showing improved accuracies over other methods.

## 1  Introduction

Bayesian nonparametric models provide flexible and accurate priors by allowing the dimensionality of the parameter space to scale with dataset sizes [12]. The Mondrian process (MP) is a Bayesian nonparametric prior for space partitioning and provides a Bayesian view of decision trees and random forests [29, 19]. Inference for the MP is conducted by recursive and random axis-aligned cuts in the domain of the observed data, partitioning the space into a hierarchical tree of hyper-rectangles.

The MP is appropriate for multi-dimensional data, and it is self-consistent (*i.e.*, it is a projective distribution), meaning that the prior distribution it induces on a subset of a domain is equal to the marginalisation of the prior over the complement of that subset. Self-consistency is required in Bayesian nonparametric models in order to insure correct inference, and prevent any bias arising from sampling and sample population size. Recent advances in MP methods for Bayesian nonparametric space partitioning include online methods [21], and particle Gibbs inference for MP additive regression trees [22]. These methods achieve high predictive accuracy, with improved efficiency. However, the axis-aligned nature of the decision boundaries of the MP restricts its flexibility, which could lead to failure in capturing inter-dimensional dependencies in the domain.

Recently, advances in Bayesian nonparametrics have been developed to allow more flexible non-axis aligned partitioning. The Ostomachion process (OP) was introduced to generalise the MP and allow non-axis aligned cuts. The OP is defined for two dimensional data domains [11]. In the OP, the angle

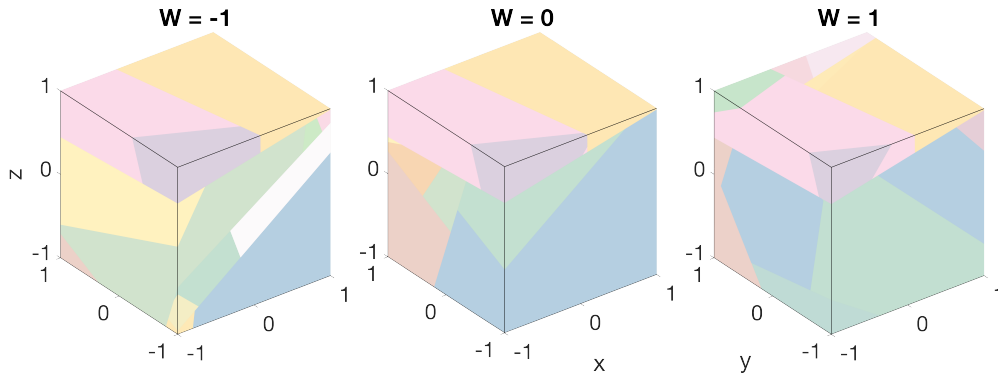

Figure 1: A draw from the uRTP prior with domain given by a four dimensional hypercube $(x, y, z, w) \in [-1, 1]^4$. Intersections of the draw and the three dimensional cube are shown for $w = -1, 0, 1$. Colours indicate polytope identity, and are randomly assigned.

and position of each cut is randomly sampled from a specified distribution. However, the OP is not self-consistent, and so the binary space partitioning-tree (BSP) process [9] was introduced to modify the cut distribution of the OP in order to recover self-consistency. The main limitation of the OP and the BSP is that they are not defined for dimensions larger than two (*i.e.*, they are restricted to data with two predictors). To relax this constraint, in [10] a self-consistent version of the BSP was extended to arbitrarily dimensioned space (called the BSP-forest). But for this process each cutting hyperplane is axis-aligned in all but two dimensions (with non-axis alignment allowed only in the remaining two dimensions, following the specification of the two dimensional BSP). Alternative constructions of non-axis aligned partitioning for two dimensional spaces and non-Bayesian methods involving sparse linear combinations of predictors or canonical correlation have also been proposed as random forest generalisations [14, 30, 28].

In this work, we propose the Random Tessellation Process (RTP), a framework for describing Bayesian nonparametric models based on cutting multi-dimensional Euclidean space. We consider four versions of the RTP, including a generalisation of the Mondrian process with non-axis aligned cuts (a sample from this prior is shown in Figure 1), a formulation of the Mondrian process as an RTP, and weighted versions of these two methods (shown in Figure 2). By virtue of their construction, all versions of the RTP are self-consistent, and are based on the theory of stable iterated tessellations in stochastic geometry [27]. The partitions induced by the RTP prior are described by a set of polytopes.

We derive a sequential Monte Carlo (SMC) algorithm [8] for RTP inference which takes advantage of the hierarchical structure of the generating process for the polytope tree. We also propose a random forest version of RTPs, which we refer to as Random Tessellation Forests (RTFs). We apply our proposed model to simulated data and several gene expression datasets, and demonstrate its effectiveness compared to other modern machine learning methods.

## 2 Methods

Suppose we observe a dataset $(\boldsymbol{v}_1, z_1), \ldots, (\boldsymbol{v}_n, z_n)$, for a classification task in which $\boldsymbol{v}_i \in \mathbb{R}^d$ are predictors and $z_i \in \{1, \ldots, K\}$ are labels (with $K$ levels, $K \in \mathbb{N}_{>1}$). Bayesian nonparametric models based on partitioning the predictors proceed by placing a prior on aspects of the partition, and associating likelihood parameters with the blocks of the partition. Inference is then done on the joint posterior of the parameters and the structure of the partition. In this section, we develop the RTP: a unifying framework that covers and extends such Bayesian nonparametric models through a prior on partitions of $(\boldsymbol{v}_1, z_1), \ldots, (\boldsymbol{v}_n, z_n)$ induced by tessellations.

### 2.1 The Random Tessellation Process

A tessellation $Y$ of a bounded domain $W \subset \mathbb{R}^d$ is a finite collection of closed polytopes such that the union of the polytopes is all of $W$, and such that the polytopes have pairwise disjoint interiors [6]. We denote tessellations of $W$ by $Y(W)$ or the symbol $\triangle$. A polytope is an intersection of finitely many closed half-spaces. In this work we will assume that all polytopes are bounded and have nonempty

interior. An RTP $Y_t(W)$ is a tessellation-valued right-continuous Markov jump process (MJP) defined on $[0, \tau]$ (we refer to the $t$-axis as time), in which events are cuts (specified by hyperplanes) of the tessellation's polytopes, and $\tau$ is a prespecified budget [2]. In this work we assume that all hyperplanes are affine (*i.e.*, they need not pass through the origin).

The initial tessellation $Y_0(W)$ contains a single polytope given by the convex hull of the observed predictors in the dataset: $W = \text{hull}\{\boldsymbol{v}_1, \ldots, \boldsymbol{v}_n\}$ (the operation hull $A$ denotes the convex hull of the set $A$). In the MJP for the random tessellation process, each polytope has an exponentially distributed lifetime, and at the end of a polytope's lifetime, the polytope is replaced by two new polytopes. The two new polytopes are formed by drawing a hyperplane that intersects the interior of the old polytope, and then intersecting the old polytope with each of the two closed half-spaces bounded by the drawn hyperplane. We refer to this operation as cutting a polytope according to the hyperplane. These cutting events continue until the prespecified budget $\tau$ is reached.

Let $H$ be the set of hyperplanes in $\mathbb{R}^d$. Every hyperplane $h \in H$ can be written uniquely as the set of points $\{P : \langle \vec{n}, P - u\,\vec{n} \rangle = 0\}$, such that $\vec{n} \in S^{d-1}$ is a normal vector of $h$, and $u \in \mathbb{R}_{\geq 0}$ ($u \geq 0$). Here $S^{d-1}$ is the unit $(d-1)$-sphere (*i.e.*, $S^{d-1} = \{\vec{n} \in \mathbb{R}^d : \|\vec{n}\| = 1\}$). Thus, there is a bijection $\varphi : S^{d-1} \times \mathbb{R}_{\geq 0} \longmapsto H$ by $\varphi(\vec{n}, u) = \{P : \langle \vec{n}, P - u\,\vec{n} \rangle = 0\}$, and therefore a measure $\Lambda$ on $H$ is induced by any measure $\Lambda \circ \varphi$ on $S^{d-1} \times \mathbb{R}_{\geq 0}$ through this bijection [20, 6].

In [27] *Section* 2.1, Nagel and Weiss describe a random tessellation associated with a measure $\Lambda$ on $H$ through a tessellation-valued MJP $Y_t$ such that the rate of the exponential distribution for the lifetime of a polytope $a \in Y_t$ is $\Lambda([a])$ (here and throughout this work, $[a]$ denotes the set of hyperplanes in $\mathbb{R}^d$ that intersect the interior of $a$), and the hyperplane for the cutting event for a polytope $a$ is sampled according to the probability measure $\Lambda(\cdot \cap [a])/\Lambda([a])$. We use this construction as the prior for RTPs, and describe their generative process in Algorithm 1. This algorithm is equivalent to the first algorithm listed in [27].

---

**Algorithm 1** Generative Process for RTPs

1: **Inputs:** a) Bounded domain $W$, b) RTP measure $\Lambda$ on $H$, c) prespecified budget $\tau$.
2: **Outputs:** A realisation of the Random Tessellation Process $(Y_t)_{0 \leq t \leq \tau}$.
3: $\tau_0 \leftarrow 0$.
4: $Y_0 \leftarrow \{W\}$.
5: **while** $\tau_0 \leq \tau$ **do**
6:     Sample $\tau' \sim \text{Exp}\left(\sum_{a \in Y_{\tau_0}} \Lambda([a])\right)$.
7:     Set $Y_t \leftarrow Y_{\tau_0}$ for all $t \in (\tau_0, \min\{\tau, \tau_0 + \tau'\}]$.
8:     Set $\tau_0 \leftarrow \tau_0 + \tau'$.
9:     **if** $\tau_0 \leq \tau$ **then**
10:         Sample a polytope $a$ from the set $Y_{\tau_0}$ with probability proportional to (*w.p.p.t.*) $\Lambda([a])$.
11:         Sample a hyperplane $h$ from $[a]$ according to the probability measure $\Lambda(\cdot \cap [a])/\Lambda([a])$.
12:         $Y_{\tau_0} \leftarrow (Y_{\tau_0}/\{a\}) \cup \{a \cap h^-, a \cap h^+\}$. ($h^-$ and $h^+$ are the $h$-bounded closed half planes.)
13:     **else**
14:         **return** the tessellation-valued right-continuous MJP sample $(Y_t)_{0 \leq t \leq \tau}$.

---

### 2.1.1 Self-consistency of Random Tessellation Processes

From *Theorem 1* in [27], if the measure $\Lambda$ is invariant with respect to translation (*i.e.*, $\Lambda(A) = \Lambda(\{h + x : h \in A\})$ for all measurable subsets $A \subset H$ and $x \in \mathbb{R}^d$), and if a set of $d$ hyperplanes with orthogonal normal vectors is contained in the support of $\Lambda$, then for all bounded domains $W' \subseteq W$, $Y_t(W')$ is equal in distribution to $Y_t(W) \cap W'$. This means that self-consistency holds for the random tessellations associated with such $\Lambda$. (Here, for a hyperplane $h$, $h + x$ refers to the set $\{y + x : y \in h\}$, and for a tessellation $Y$ and a domain $W'$, $Y \cap W'$ is the tessellation $\{a \cap W' : a \in Y\}$.) In [27], such tessellations are referred to as stable iterated tessellations.

If we assume that $\Lambda \circ \varphi$ is the product measure $\lambda^d \times \lambda_+$, with $\lambda^d$ symmetric (*i.e.*, $\lambda^d(A) = \lambda^d(-A)$ for all measurable sets $A \subseteq S^{d-1}$) and further that $\lambda_+$ is given by the Lebesgue measure on $\mathbb{R}_{\geq 0}$, then $\Lambda$ is translation invariant (a proof of this statement is given in Appendix A, *Lemma 1* of the *Supplementary Material*). So, through Algorithm 1 and *Theorem 1* in [27], any distribution $\lambda^d$ on the sphere $S^{d-1}$ that is supported on a set of $d$ hyperplanes with orthogonal normal vectors gives rise to

a self-consistent random tessellation, and we refer to models based on this self-consistent prior as Random Tessellation Processes (RTPs). We refer to any product measure $\Lambda \circ \varphi = \lambda^d \times \lambda_+$ such these conditions hold (*i.e.*, $\lambda^d$ symmetric, $\Lambda$ supported on $d$ orthogonal hyperplanes and $\lambda_+$ given by the Lebesgue measure) as an RTP measure.

### 2.1.2 Relationship to cutting nonparametric models

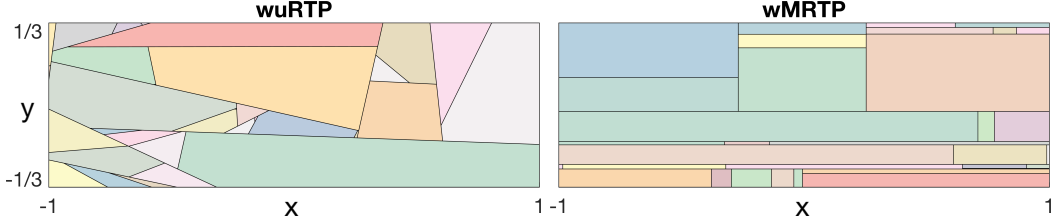

Figure 2: Draws from priors *Left)* wuRTP, and *Right)* wMRTP, for the domain given by the rectangle $W = [-1, 1] \times [-1/3, 1/3]$. Weights are given by $\omega_x = 14, \omega_y = 1$, leading to horizontal *(x-axis heavy)* structure in the polygons. Colours are randomly assigned and indicate polygon identity, and black delineates polygon boundaries.

If $\lambda^d$ is the measure associated with a *uniform distribution* on the sphere (with respect to the usual Borel sets on $S^{d-1}$ [18]), then the resulting RTP is a generalisation of the Mondrian process with non-axis aligned cuts. We refer to this RTP as the uRTP (for uniform RTP). In this case, $\lambda^d$ is a probability measure and a normal vector $\vec{n}$ may be sampled according to $\lambda^d$ by sampling $n_i \sim N(0, 1)$ and then setting $\vec{n}_i = n_i / \|n\|$. A draw from the uRTP prior supported on a four dimensional hypercube is displayed in Figure 1.

We consider a weighted version of the uniform RTP found by setting $\lambda^d$ to the measure associated with the distribution on $\vec{n}$ induced by the scheme $n_i \sim N(0, \omega_i^2)$, $\vec{n}_i = n_i / \|n\|$. We refer to this RTP as the wuRTP (weighted uniform RTP), and the wuRTP is parameterised by $d$ weights $\omega_i \in \mathbb{R}_{>0}$. Note that the isotropy of the multivariate Gaussian $n$ implies symmetry of $\lambda^d$. Setting the weight $\omega_i$ increases the prior probability of cuts orthogonal to the $i$-th predictor dimension, allowing prior information about the importance of each of the predictors to be incorporated. Figure 2*(left)* shows a draw from the wuRTP prior supported on a rectangle.

The Mondrian process itself is an RTP with $\lambda^d = \sum_{v \in \text{poles}(d)} \delta^v$. Here, $\delta_x$ is the Dirac delta supported on $x$, and poles$(d)$ is the set of normal vectors with zeros in all coordinates, except for one of the coordinates (the non-zero coordinate of these vectors can be either $-1$ or $+1$). We refer to this view of the Mondrian process as the MRTP. If $\lambda^d = \sum_{v \in \text{poles}(d)} \omega_{i(v)} \delta_v$, where $\omega_i$ are $d$ axis weights, and $i(v)$ is the index of the nonzero element of $v$, then we arrive at a weighted version of the MRTP, which we refer to as the wMRTP. Figure 2*(right)* displays a draw from the wMRTP prior. The horizontal organisation of the lines in Figure 2 arise from the uneven axis weights.

Other nonparametric models based on cutting polytopes may also be viewed in this way. For example, Binary Space Partitioning-Tree Processes [9] are uRTPs and wuRTPs restricted to two dimensions, and the generalization of the Binary Space Partitioning-Tree Process in [10] is an RTP for which $\lambda^d$ is a sum of delta functions convolved with smooth measures on $S^1$ projected onto pairs of axes. The Ostomachion process [11] does not arise from an RTP: it is not self-consistent, and so by *Theorem 1* in [27], the OP cannot arise from an RTP measure.

### 2.1.3 Likelihoods for Random Tessellation Processes

In this section, we illustrate how RTPs can be used to model categorical data. For example, in gene expression data of tissues, the predictors are the amounts of expression of each gene in the tissue (the vector $\boldsymbol{v}_i$ for sample $i$), and our goal is to predict disease condition (labels $z_i$). Let $\triangle_t$ be an RTP on the domain $W = \text{hull}\{\boldsymbol{v}_1, \dots, \boldsymbol{v}_n\}$. Let $J_t$ denote the number of polytopes in $\triangle_t$. We let $h(\boldsymbol{v}_i)$ denote a mapping function, which matches the $i$-th data item to the polytope in the tessellation containing that data item. Hence, $h(\boldsymbol{v}_i)$ takes a value in the set $\{1, \dots, J_t\}$. We will consider the

likelihood arising at time $t$ from the following generative process:

$$\triangle_t \sim \boldsymbol{Y}_t(\boldsymbol{W}), \qquad \boldsymbol{\phi}_j \sim \text{Dirichlet}(\boldsymbol{\alpha}) \text{ for } 1 \leq j \leq J_t, \tag{1}$$

$$z_i | \triangle_t, \boldsymbol{\phi}_{h(\boldsymbol{v}_i)} \sim \text{Multinomial}(\boldsymbol{\phi}_{h(\boldsymbol{v}_i)}) \text{ for } 1 \leq i \leq n. \tag{2}$$

Here $\boldsymbol{\phi}_j = (\phi_{j1}, \ldots, \phi_{jK})$ are parameters of the multinomial distribution with a Dirichlet prior with hyperparameters $\boldsymbol{\alpha} = (\alpha_k)_{1 \leq k \leq K}$. The likelihood function for $\boldsymbol{Z} = (z_i)_{1 \leq i \leq n}$ conditioned on the tessellation $\triangle_t$ and $\boldsymbol{V} = (v_i)_{1 \leq i \leq n}$ and given the hyperparameter $\boldsymbol{\alpha}$ is as follows:

$$P(\boldsymbol{Z} | \triangle_t, \boldsymbol{V}, \boldsymbol{\alpha}) = \int \cdots \int P(\boldsymbol{Z}, \phi | \triangle_t, \boldsymbol{\alpha}) d\boldsymbol{\phi}_1 \cdots d\boldsymbol{\phi}_{J_t} = \prod_{j=1}^{J_t} \frac{B(\boldsymbol{\alpha} + \boldsymbol{m}_j)}{B(\boldsymbol{\alpha})}. \tag{3}$$

Here $B(\cdot)$ is the multivariate beta function, $\boldsymbol{m}_j = (m_{jk})_{1 \leq k \leq K}$ and $m_{jk} = \sum_{i:h(\boldsymbol{v}_i)=j} \delta(z_i = k)$, and $\delta(\cdot)$ is an indicator function with $\delta(z_i = k) = 1$ if $z_i = k$ and $\delta(z_i = k) = 0$ otherwise. We refer to Appendix A, *Lemma 2* of the *Supplementary Material* for the derivation of (3).

## 2.2   Inference for Random Tessellation Processes

Our objective is to infer the posterior of the tessellation at time $t$, denoted $\pi(\triangle_t | \boldsymbol{V}, \boldsymbol{Z}, \boldsymbol{\alpha})$. We let $\pi_0(\triangle_t)$ denote the prior distribution of $\triangle_t$. Let $P(\boldsymbol{Z} | \triangle_t, \boldsymbol{V}, \boldsymbol{\alpha})$ denote the likelihood (3) of the data given the tessellation $\triangle_t$ and the hyperparameter $\boldsymbol{\alpha}$. By Bayes' rule, the posterior of the tessellation at time $t$ is $\pi(\triangle_t | \boldsymbol{V}, \boldsymbol{Z}, \boldsymbol{\alpha}) = \pi_0(\triangle_t) P(\boldsymbol{Z} | \triangle_t, \boldsymbol{V}, \boldsymbol{\alpha}) / P(\boldsymbol{Z} | \boldsymbol{V}, \boldsymbol{\alpha})$.

Here $P(\boldsymbol{Z} | \boldsymbol{V}, \boldsymbol{\alpha})$ is the marginal likelihood given data $\boldsymbol{Z}$. This posterior distribution is intractable, and so in *Section* 2.2.3 we introduce an efficient SMC algorithm for conducting inference on $\pi(\triangle_t)$. The proposal distribution for this SMC algorithm involves draws from the RTP prior, and so in *Section* 2.2.1, we describe a rejection sampling scheme for drawing a hyperplane from the probability measure $\Lambda(\cdot \cap [a]) / \Lambda([a])$ for a polytope $a$. We provide some optimizations and approximations used in this SMC algorithm in *Section* 2.2.2.

### 2.2.1   Sampling cutting hyperplanes for Random Tessellation Processes

Suppose that $a$ is a polytope, and $\Lambda \circ \varphi$ is an RTP measure such that $\Lambda(\varphi(\cdot)) = (\lambda^d \times \lambda_+)(\cdot)$. We wish to sample a hyperplane according to the probability measure $\Lambda(\cdot \cap [a]) / \Lambda([a])$. We note that if $B_r(x)$ is the smallest closed $d$-ball containing $a$ (with radius $r$ and centre $x$), then $[a]$ is contained in $[B_r(x)]$. If we can sample a normal vector $\vec{n}$ according to $\lambda^d$, then we can sample a hyperplane according to $\Lambda(\cdot \cap [a]) / \Lambda([a])$ through the following rejection sampling scheme.

- Step 1) Sample $\vec{n}$ according to $\lambda^d$.
- Step 2) Sample $u \sim \text{Uniform}[0, r]$.
- Step 3) If the hyperplane $h = x + \{P : \langle \vec{n}, P - u\,\vec{n} \rangle\}$ intersects $a$, then `RETURN` $h$. Otherwise, `GOTO` Step 1).

Note that in Step 3, the set $\{P : \langle \vec{n}, P - u\,\vec{n} \rangle\}$ is a hyperplane intersecting the ball $B_r(0)$ centred at the origin, and so translation of this hyperplane by $x$ yields a hyperplane intersecting the ball $B_r(x)$. When this scheme is applied to the uRTP or wuRTP, in Step 1 $\vec{n}$ is sampled from the uniform distribution on the sphere or the appropriate isotropic Gaussian. And for the MRTP or wMRTP, $\vec{n}$ is sampled from the discrete distributions given in *Section* 2.1.2.

### 2.2.2   Optimizations and approximations

We use three methods to decrease the computational requirements and complexity of inference based on RTP posteriors. First, we replace all polytopes with convex hulls formed by intersecting the polytopes with the dataset predictors. Second, in determining the rates of the lifetimes of polytopes, we approximate $\Lambda([a])$ with the measure $\Lambda([\cdot])$ applied to the smallest closed ball containing $a$. Third, we use a pausing condition so that no cuts are proposed for polytopes for which the labels of all predictors in that polytope are the same label.

**Convex hull replacement**. In our posterior inference, if $a \in Y$ is cut according to the hyperplane $h$, we consider the resulting tessellation to be $Y/\{a\} \cup \{\text{hull}(a \cap \boldsymbol{V} \cap h^+), \text{hull}(a \cap \boldsymbol{V} \cap h^-)\}$. Here $h^+$

and $h^-$ are the two closed half planes bounded by $h$, and $/$ is the set minus operation. This requires a slight loosening of the definition of a tessellation $Y$ of a bounded domain $W$ to allow the union of the polytopes of a tessellation $Y$ to be a strict subset of $W$ such that $\boldsymbol{V} \subseteq \cup_{a \in Y} a$.

In our computations, we do not need to explicitly compute these convex hulls, and instead for any polytope $b$, we store only $b \cap \boldsymbol{V}$, as this is enough to determine whether or not a hyperplane $h$ intersects $\text{hull}(b \cap \boldsymbol{V})$. This membership check is the only geometric operation required to sample hyperplanes intersecting $b$ according to the rejection sampling scheme from *Section* 2.2.1. By the self-consistency of RTPs, this has the effect of marginalizing out MJP events involving cuts that do not further separate the predictors in the dataset. This also obviates the need for explicit computation of the facets of polytopes, significantly simplifying the codebase of our implementation of inference.

After this convex hull replacement operation, a data item in the test dataset may not be contained in any polytope, and so to conduct posterior inference we augment the training dataset with a version of the testing dataset in which the label is missing, and then marginalise the missing label in the likelihood described in *Section* 2.1.3.

**Spherical approximation**. Every hyperplane intersecting a polytope $a$ also intersects a closed ball containing $a$. Therefore, for any RTP measure $\Lambda$, $\Lambda([a])$ is upper bounded by $\Lambda([B(r_a)])$. Here $r_a$ is the radius of the smallest closed ball containing $a$. We approximate $\Lambda([a]) \simeq \Lambda([B(r_a)])$ for use in polytope lifetime calculations in our uRTP inference and we do not compute $\Lambda([a])$ exactly. For the uRTP and wRTP, $\Lambda([B(r_a)]) = r_a$. A proof of this is given in Appendix A, *Lemma 3* of the *Supplementary Material*. For the MRTP and wMRTP, $\Lambda([a])$ can be computed exactly [29].

**Pausing condition**. In our posterior inference, if $z_i = z_j$ for all $i, j$ such that $\boldsymbol{v}_i, \boldsymbol{v}_j \in a$, then we *pause* the polytope $a$ and no further cuts are performed on this polytope. This improves computational efficiency without affecting inference, as cutting such a polytope cannot further separate labels. This was done in recent work for Mondrian processes [21] and was originally suggested in the Random Forest reference implementation [4].

### 2.2.3 Sequential Monte Carlo for Random Tessellation Process inference

---

**Algorithm 2** SMC for inferring RTP posteriors

---

1: **Inputs:** a) Training dataset $\boldsymbol{V}$, $\boldsymbol{Z}$, b) RTP measure $\Lambda$ on $H$, c) prespecified budget $\tau$, d) likelihood hyperparameter $\alpha$.
2: **Outputs:** Approximate RTP posterior $\sum_{m=1}^M \varpi_m \delta_{\triangle_{\tau,m}}$ at time $\tau$. ($\varpi_m$ are particle weights.)
3: Set $\tau_m \leftarrow 0$, for $m = 1, \ldots, M$.
4: Set $\triangle_{0,m} \leftarrow \{\text{hull } \boldsymbol{V}\}$, $\varpi_m \leftarrow 1/M$, for $m = 1, \ldots, M$.
5: **while** $\min\{\tau_m\}_{m=1}^M < \tau$ **do**
6:     Resample $\triangle'_{\tau_m,m}$ from $\{\triangle_{\tau_m,m}\}_{m=1}^M$ *w.p.p.t.* $\{\varpi_m\}_{m=1}^M$, for $m = 1, \ldots, M$.
7:     Set $\triangle_{\tau_m,m} \leftarrow \triangle'_{\tau_m,m}$, for $m = 1, \ldots, M$.
8:     Set $\varpi_m \leftarrow 1/M$, for $m = 1, \ldots, M$.
9:     **for** $m \in \{m : m = 1, \ldots, M \text{ and } \tau_m < \tau\}$ **do**
10:         Sample $\tau' \sim \text{Exp}\left(\sum_{a \in \triangle_{\tau_m,m}} r_a\right)$. ($r_a$ is the radius of the smallest closed ball containing $a$.)
11:         Set $\triangle_{t,m} \leftarrow \triangle_{\tau_m,m}$, for all $t \in (\tau_m, \min\{\tau, \tau_m + \tau'\}]$.
12:         **if** $\tau_m + \tau' \leq \tau$ **then**
13:             Sample $a$ from the set $\triangle_{\tau_m,m}$ *w.p.p.t.* $r_a$.
14:             Sample $h$ from $[a]$ according to $\Lambda(\cdot \cap [a])/\Lambda([a])$ using *Section* 2.2.1.
15:             Set $\triangle_{\tau_m,m} \leftarrow (\triangle_{\tau_m,m}/\{a\}) \cup \{\text{hull}(\boldsymbol{V} \cap a \cap h^-), \text{hull}(\boldsymbol{V} \cap a \cap h^+)\}$.
16:             Set $\varpi_m \leftarrow \varpi_m P(\boldsymbol{Z}|\triangle_{\tau_m,m}, \boldsymbol{V}, \alpha)/P(\boldsymbol{Z}|\triangle'_{\tau_m,m}, \boldsymbol{V}, \alpha)$ according to (3).
17:         **else**
18:             Set $\triangle_{t,m} \leftarrow \triangle_{\tau_m,m}$, for $t \in (\tau_m, \tau]$.
19:         Set $\tau_m \leftarrow \tau_m + \tau'$.
20:     Set $\mathcal{Z} \leftarrow \sum_{m=1}^M \varpi_m$.
21:     Set $\varpi_m \leftarrow \varpi_m/\mathcal{Z}$, for $m = 1, \ldots, M$.
22: **return** the particle approximation $\sum_{m=1}^M \varpi_m \delta_{\triangle_{\tau,m}}$.

---

We provide an SMC method (Algorithm 2) with $M$ particles, to approximate the posterior distribution $\pi(\triangle_t)$ of an RTP conditioned on $\boldsymbol{Z}, \boldsymbol{V}$, and given an RTP measure $\Lambda$ and a hyperparameter $\boldsymbol{\alpha}$ and a prespecified budget $\tau$. Our algorithm iterates between three steps: resampling particles (Algorithm 2, line 7), propagation of particles (Algorithm 2, lines 13-15) and weighting of particles (Algorithm 2, line 16). At each SMC iteration, we sample the next MJP events using the spherical approximation of $\Lambda([\cdot])$ described in *Section* 2.2.2. For brevity, the pausing condition described in *Section* 2.2.2 is omitted from Algorithm 2.

In our experiments, after using Algorithm 2 to yield a posterior estimate $\sum_{m=1}^{M} \varpi_m \delta_{\triangle_{\tau,m}}$, we select the tessellation $\triangle_{\tau,m}$ with the largest weight $\varpi_m$ (*i.e.*, we do not conduct resampling at the last SMC iteration). We then compute posterior probabilities of the test dataset labels using the particle $\triangle_{\tau,m}$. This method of not resampling after the last SMC iteration is recommended in [7] for lowering asymptotic variance in SMC estimates.

The computational complexity of Algorithm 2 depends on the number of polytopes in the tessellations, and the organization of the labels within the polytopes. The more linearly separable the dataset is, the sooner the pausing conditions are met. The complexity of computing the spherical approximation in *Section* 2.2.2 (the radius $r_a$) for a polytope $a$ is $\mathcal{O}(|\boldsymbol{V} \cap a|^2)$, where $|\cdot|$ denotes set cardinality.

### 2.2.4 Prediction with Random Tessellation Forests

Random forests are commonly used in machine learning for classification and regression problems [3]. A random forest is represented by an ensemble of decision trees, and predictions of test dataset labels are combined over all decision trees in the forest. To improve the performance of our methods, we consider random forest versions of RTPs (which we refer to as RTFs: uRTF, wuRTF, MRTF, wMRTF are random forest versions of the uRTP, MRTP and their weighted versions *resp.*). We run Algorithm 2 independently $T$ times, and predict labels using the modes. Differing from [3], we do not use bagging.

In [22], Lakshminarayanan, Roy and Teh consider an efficient Mondrian forest in which likelihoods are dropped from the SMC sampler and cutting is done independent of likelihood. This method follows recent theory for random forests [15]. We consider this method (by dropping line 16 of Algorithm 2) and refer to the implementations of this method as the uRTF.i and MRTF.i (*i* for likelihood *i*ndependence).

## 3 Experiments

In *Section* 3.1, we explore a simulation study that shows differences among uRTP and MRTP, and some standard machine learning methods. Variations in gene expression across tissues in brain regions play an important role in disease conditions. In *Section* 3.2, we examine predictions of a variety of RTF models for gene expression data. For all our experiments, we set the likelihood hyperparameters for the RTPs and RTFs to the empirical estimates $\alpha_k$ to $n_k/1000$. Here $n_k = \sum_{i=1}^{n} \delta(z_i = k)$. In all of our experiments, for each train/test split, we allocate 60% of the data items at random to the training set.

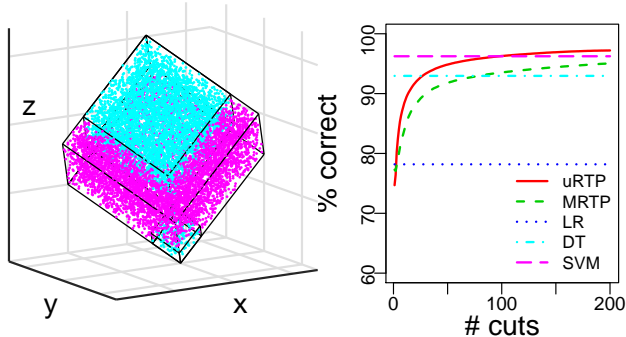

Figure 3: *Left)* A view of the Mondrian cube, with cyan indicating label 1, magenta indicating label 2, and black delineating label boundaries. *Right)* Percent correct versus number of cuts for predicting Mondrian cube test dataset, with uRTP, MRTP and a variety of baseline methods.

An implementation of our methods (released under the open source BSD 2-clause license) and a software manual are provided in the *Supplementary Material*.

## 3.1 Simulations on the Mondrian cube

We consider a simulated three dimensional dataset designed to exemplify the difference between axis-aligned and non-axis aligned models. We refer to this dataset as the Mondrian cube, and we investigate the performance of uRTP and the MRTP on this dataset, along with some standard machine learning approaches, varying the number of cuts in the processes. The Mondrian cube dataset is simulated as follows: first, we sample 10,000 points uniformly in the cube $[0, 1]^3$. Points falling in the cube $[0, 0.25]^3$ or the cube $[0.25, 1]^3$ are given label 1, and the remaining points are given label 2. Then, we centre the points and rotate all of the points by the angles $\frac{\pi}{4}$ and $-\frac{\pi}{4}$ about the $x$-axis and $y$-axis respectively, creating a dataset organised on diagonals. In Figure 3*(left)*, we display a visualization of the Mondrian cube dataset, wherein points are colored by their label. We apply the SMC algorithm to the Mondrian cube data, with 50 random train/test splits. For each split, we run 10 independent copies of the uRTP and MRTP and take the mode of their results, and we also examine the accuracy of logistic regression (LR), a decision tree (DT) and a support vector machine (SVM).

Figure 3*(right)* shows that the percent correct for the uRTP and MRTP both increase as the number of cuts increases, and plateaus when the number of cuts becomes larger (greater than 25). Even though the uRTP has lower accuracy at the first cut, it starts dominating the MRTP after the second cut. Overall, in terms of percent correct, with any number of cuts $> 105$, a sign test indicates that the uRTP performs significant better than all other methods at nominal significance, and the MRTP performs significant better than DT and LR for any number of cuts $> 85$ at nominal significance.

## 3.2 Experiment on gene expression data in brain tissue

We evaluate a variety of RTPs and some standard machine learning methods on a glioblastoma tissue dataset *GSE83294* [13], which includes 22,283 gene expression profiles for 85 astrocytomas (26 diagnosed as grade III and 59 as grade IV). We also examine schizophrenia brain tissue datasets: *GSE21935* [1], in which 54,675 gene expression in the superior temporal cortex is recorded for 42 subjects, with 23 cases (with schizophrenia), and 19 controls (without schizophrenia), and dataset *GSE17612* [24], a collection of 54,675 gene expressions from samples in the anterior prefrontal cortex (*i.e.*, a different brain area from *GSE21935*) with 28 schizophrenic subjects, and 23 controls. We refer to these datasets as *GL85*, *SCZ42* and *SCZ51*, respectively.

We also consider a combined version of *SCZ42* and *SCZ51* (in which all samples are concatenated), which we refer to as *SCZ93*. For *GL85* the labels are the astrocytoma grade, and for *SCZ42*, *SCZ51* and *SCZ93* the labels are schizophrenia status. We use principal components analyais (PCA) in preprocessing to replace the predictors of each data item (a set of gene expressions) with its scores on a full set of principal components (PCs): *i.e.*, 85 PCs for *GL85*, 42 PCs in *SCZ42* and 51 PCs in *SCZ51*. We then scale the PCs. We consider 200 test/train splits for each dataset. These datasets were acquired from NCBI's Gene Expression Omnibus[1] and were are released under the *Open Data Commons Open Database License*. We provide test/train splits of the PCA preprocessed datasets in the *Supplementary Material*.

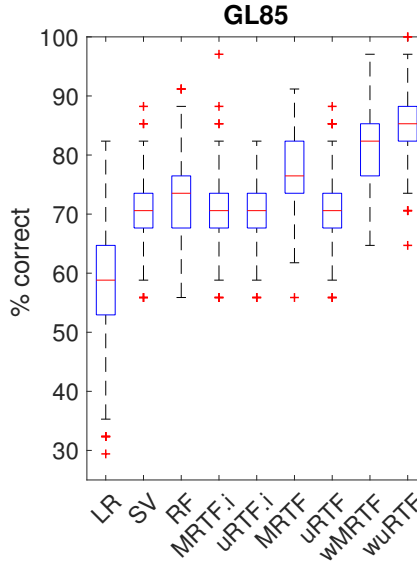

Figure 4: Box plot showing wuRTF and wMRTF improvements for *GL85*, and generally best performance for wuRTF method (with sign test $p$-value of $3.2 \times 10^{-9}$ *vs* wMRTF). Reduced performance of SVM indicates structure in *GL85* that is not linearly separable. Medians, quantiles and outliers beyond 99.3% coverage are indicated.

Through this preprocessing, the $j$-th predictor is the score vector of the $j$-th principal component. For the weighted RTFs (the wuRTF and wMRTF), we set the weight of the $j$-th predictor to be proportional to the variance explained by the $j$-th PC ($\sigma_j^2$): $\omega_j = \sigma_j^2$. We set the number of trees in

| Dataset | BL | LR | SVM | RF | MRTF.i | uRTF.i | MRTF | uRTF | wMRTF | wuRTF |
|---------|-----|-----|-----|-----|--------|--------|------|------|-------|-------|
| *GL85*  | 70.34 | 58.13 | 70.34 | 73.01 | 70.74 | 70.06 | 77.09 | 70.60 | 80.57 | **84.90** |
| *SCZ42* | 46.68 | **57.65** | 46.79 | 51.76 | 49.56 | 48.50 | 49.91 | 47.71 | **53.12** | **53.97** |
| *SCZ51* | 46.55 | 51.15 | 46.67 | **57.38** | 52.55 | 48.58 | **57.95** | 44.70 | **58.12** | 49.05 |
| *SCZ93* | 48.95 | **53.05** | 50.15 | 52.45 | 50.23 | 50.24 | 51.80 | 50.34 | **53.12** | **54.99** |

Table 1: Comparison of mean percent correct on gene expression datasets over 200 random train/test splits across different methods. Nominal statistical significance ($p$-value $< 0.05$) is ascertained by a sign test in which ties are broken in a conservative manner. Tests are performed between the top method and each other method. Bold values indicate the top method and all methods statistically indistinguishable from the top method according to nominal significance. Largest nominally significant improvement is seen for wuRTF on *GL85*, and wuRTF is significantly better than other methods for this dataset. The wMRTF and wuRTF have largest mean percent correct for *SCZ51* and *SCZ93* but are not statistically distinguishable from *RF* or *LR* for those datasets.

all of the random forests to 100, which is the default in R's `randomForest` package [23]. For the all RTFs, we set the budget $\tau = \infty$, as is done in [21].

## 4 Results

We compare percent correct for the wuRTF, uRTF, uRTF.i, and the Mondrian Random Tessellation Forests wMRTF, MRTF and MRTF.i, a random forest (RF), logistic regression (LR), a support vector machine (SVM) and a baseline (BL) in which the mode of the training set label is always predicted [25, 23]. Mean percent correct and sign tests for all of these experiments are reported in Table 1 and box plots for the *GL85* experiment reported in Figure 4. We observe that the increase in accuracy of wuRTFs achieves nominal significance over other methods on *GL85*. For datasets *SCZ42*, *SCZ51* and *SCZ93*, the performance of the RTFs is comparable to that of logistic regression and random forests. For all datasets we consider, RTFs have higher accuracy than SVMs (with nominal significance). Boxplots with accuracies for the datasets *SCZ42*, *SCZ51* and *SCZ93* are provided in Appendix B, Supplementary Figure 1 of the *Supplementary Material*. Results of a conservative pairwise sign test performed between each pair of methods on each dataset, standard deviation of the percent correct, and mean runtime across different methods for all these four datasets are reported in Supplementary Tables 1, 2, and 3 in Appendix B of the *Supplementary Material*.

## 5 Discussion

The spherical approximation introduced in Section 2.2.2 can lead to inexact inference. In Supplementary Algorithm 1, we introduce a new algorithm based on Poisson thinning that recovers exact inference. This algorithm may improve accuracy and allow hierarchical likelihoods.

There are many directions for future work in RTPs including improved SMC sampling for MJPs as in [17], hierarchical likelihoods and online methods as in [21], analysis of minimax convergence rates [26], and extensions using Bayesian additive regression trees [5, 22]. We could also consider applications of RTPs to data that naturally displays tessellation and cracking, such as sea ice [16].

## 6 Conclusion

We have described a framework for viewing Bayesian nonparametric methods based on space partitioning as Random Tessellation Processes. This framework includes the Mondrian process as a special case, and includes extensions of the Mondrian process allowing non-axis aligned cuts in high dimensional space. The processes are self-consistent, and we derive inference using sequential Monte Carlo and random forests. To our knowledge, this is the first work to provide self-consistent Bayesian nonparametric hierarchical partitioning with non-axis aligned cuts that is defined for more than two dimensions. As demonstrated by our simulation study and experiments on gene expression data, these non-axis aligned cuts can improve performance over the Mondrian process and other machine learning methods such as support vector machines, and random forests.

**Acknowledgments**

We are grateful to Kevin Sharp, Frauke Harms, Maasa Kawamura, Ruth Van Gurp, Lars Buesing, Tom Loughin and Hugh Chipman for helpful discussion, comments and inspiration. We would also like to thank Fred Popowich and Martin Siegert for help with computational resources at Simon Fraser University. YWT's research leading to these results has received funding from the European Research Council under the European Union's Seventh Framework Programme (FP7/2007-2013) ERC grant agreement no. 617071. This research was also funded by NSERC grant numbers RGPIN/05484-2019, DGECR/00118-2019 and RGPIN/06131-2019.

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
