[Supplementary Material · appendices.pdf]

# Random Tessellation Forests: Appendices

**Shufei Ge**[1]
shufei_ge@sfu.ca

**Shijia Wang**[2,1]
shijia_wang@sfu.ca

**Yee Whye Teh**[3]
y.w.teh@stats.ox.ac.uk

**Liangliang Wang**[1]
liangliang_wang@sfu.ca

**Lloyd T. Elliott**[1]
lloyd_elliott@sfu.ca

[1]Department of Statistics and Actuarial Science, Simon Fraser University
[2]School of Statistics and Data Science, LPMC & KLMDASR, Nankai University, China
[3]Department of Statistics, University of Oxford

## Appendix A: Theorems and proofs

**Lemma 1**. *With the notation established in* Section *2.1 of the main text, let* $\Lambda \circ \varphi$ *be a product measure* $\lambda^d \times \lambda_+$ *on* $S^{d-1} \times \mathbb{R}_{\geq 0}$. *If* $\lambda^d$ *is symmetric and if* $\lambda_+$ *is the Lebesgue measure, then* $\Lambda$ *is a translation invariant measure on the set* $H$ *of hyperplanes in* $\mathbb{R}^d$.

*Proof:* We will first show that $\Lambda$ is also symmetric (*i.e.*, invariant under reflection through the origin). Suppose that $A \subseteq H$ is a $\Lambda$-measurable set of hyperplanes.

$$\Lambda(A) = (\lambda^d \times \lambda_+)(\varphi^{-1}A) \tag{1}$$

$$= \int_0^\infty \lambda^d \left( \{\vec{n} : (\vec{n}, u) \in \varphi^{-1}A\} \right) du \tag{2}$$

$$= \int_0^\infty \lambda^d \left( \{-\vec{n} : (\vec{n}, u) \in \varphi^{-1}A\} \right) du = \Lambda(-A). \tag{3}$$

We use the symmetry of $\lambda^d$ in (3), and in (2) we use the definition of product measures [1].

Let $x \in \mathbb{R}^d$. If $h \in A$, and $\varphi^{-1}h = (\vec{n}, u)$, then $h = \{P : \langle \vec{n}, P - u\vec{n} \rangle = 0\}$ and $h + x = \{P : \langle \vec{n}, P - (u + \langle x, \vec{n} \rangle) \vec{n} \rangle = 0\}$. Let $A^+ + x = \{h + x \in A + x : \varphi^{-1}h = (\vec{n}, u) \text{ and } u + \langle x, \vec{n} \rangle \geq 0\}$ and let $A^- + x = \{h + x \in A + x : \varphi^{-1}h = (\vec{n}, u) \text{ and } u + \langle x, \vec{n} \rangle < 0\}$. By these definitions, $(A^+ + x) \cup (A^- + x) = A + x$.

$$\Lambda(A + x) = \Lambda(A^+ + x) + \Lambda(A^- + x) = \Lambda(A^+ + x) + \Lambda(-(A^- + x)) \tag{4}$$

$$= \int_{\vec{n} \in S^{d-1}} \lambda_+(\{u : (\vec{n}, u) \in \varphi^{-1}(A^+ + x)\}) + \lambda_+(\{u : (\vec{n}, u) \in \varphi^{-1} -(A^- + x)\}) d\lambda^d(\vec{n}) \tag{5}$$

$$= \int_{\vec{n} \in S^{d-1}} \lambda_+(\{u + \langle x, \vec{n} \rangle : (\vec{n}, u) \in \varphi^{-1}A^+\}) + \lambda_+(\{u + \langle x, \vec{n} \rangle : (\vec{n}, u) \in \varphi^{-1} -A^-\}) d\lambda^d(\vec{n}) \tag{6}$$

$$= \int_{\vec{n} \in S^{d-1}} \lambda_+(\{u : (\vec{n}, u) \in A^+\}) + \lambda_+(\{u : (\vec{n}, u) \in -A^-\}) d\lambda^d(\vec{n}) \tag{7}$$

$$= \Lambda(A^+) + \Lambda(-A^-) = \Lambda(A^+) + \Lambda(A^-) = \Lambda(A). \tag{8}$$

Thus, $\Lambda$ is translation invariant. In (4) and (8), we use the symmetry of $\Lambda$ (from the first paragraph of this proof). In (7), we use the translation invariance of $\lambda_+$ and in (5), we use the definition of product measures [1]. In (6), we use the definitions of $A^+ + x$ and $A^- + x$. Note that $(\vec{n}, u) \in A^+ + x \Leftrightarrow (\vec{n}, u + \langle x, \vec{n} \rangle) \in A^+$ and $(\vec{n}, u) \in -(A^- + x) \Leftrightarrow (\vec{n}, u + \langle x, \vec{n} \rangle) \in -A^-$.

The intuition behind this proof is that translation by any $x$ in $A$ induces a measure preserving shear in $\Lambda \circ \varphi$, as in the stochastic geometry chapter of [2] (this is shown for $d = 2$ in that chapter). In addition, the problem that arises from the fact that translation of a hyperplane through the origin in $H$ flips the sign of $\vec{n}$ is handled by lifting the symmetry of $\lambda^d$ in $S^{d-1}$ to symmetry of $\Lambda$, which requires the decomposition of $A + x$ into a subsets with and without the sign flip (the subsets $A^+ + x$, and $A^- + x$ *resp.*).

**Lemma 2**. The likelihood function of the labels $\mathbf{Z}$ given the tessellation $\triangle_t$ and the labels $\mathbf{V}$ and the hyperparameter $\boldsymbol{\alpha}$ can be computed as follows:

$$P(\mathbf{Z}|\triangle_t, \mathbf{V}, \boldsymbol{\alpha}) = \int \cdots \int P(\mathbf{Z}, \{\boldsymbol{\phi}_j\}_{1 \leq j \leq J_t}|\triangle_t, \boldsymbol{\alpha})\, d\boldsymbol{\phi}_1 \cdots d\boldsymbol{\phi}_{J_t} \tag{9}$$

$$= \int \cdots \int \prod_{j=1}^{J_t} P(\boldsymbol{\phi}_j) \prod_{i=1}^{n} P(z_i|\triangle_t, \boldsymbol{\phi}_{h(\boldsymbol{v}_i)})\, d\boldsymbol{\phi}_1 \cdots d\boldsymbol{\phi}_{J_t} \tag{10}$$

$$= \int \cdots \int \prod_{j=1}^{J_t} P(\boldsymbol{\phi}_j) \prod_{i:h(\boldsymbol{v}_i)=j,\, k=1}^{K} \phi_{jk}^{\delta(z_i=k)}\, d\boldsymbol{\phi}_1 \cdots d\boldsymbol{\phi}_{J_t}. \tag{11}$$

In (9), we use the conditional independence between $\mathbf{Z}$ and $\triangle_t$. Let $m_{jk} = \sum_{i:h(\boldsymbol{v}_i)=j} \delta(z_i = k)$, and let $\boldsymbol{m}_j = (m_{jk})_{1 \leq k \leq K}$.

$$P(\mathbf{Z}|\triangle_t) = \int \cdots \int \prod_{j=1}^{J_t} P(\boldsymbol{\phi}_j) \prod_{k=1}^{K} \phi_{jk}^{m_{jk}}\, d\boldsymbol{\phi}_1 \cdots d\boldsymbol{\phi}_{J_t} \tag{12}$$

$$= \int \cdots \int \prod_{j=1}^{J_t} \frac{1}{B(\boldsymbol{\alpha})} \prod_{k=1}^{K} \phi_{jk}^{\alpha_k - 1} \prod_{k=1}^{K} \phi_{jk}^{m_{jk}}\, d\boldsymbol{\phi}_1 \cdots d\boldsymbol{\phi}_{J_t} \tag{13}$$

$$= \prod_{j=1}^{J_t} \frac{B(\boldsymbol{\alpha} + \boldsymbol{m}_j)}{B(\boldsymbol{\alpha})}. \tag{14}$$

Here $B(\cdot)$ is the multivariate beta function [3].

**Lemma 3**. *In the notation from* Section *2.1 of the main text, if $\Lambda$ is the uRTP measure such that $\Lambda \circ \varphi = \lambda^d \times \lambda_+$ and $\lambda^d$ is measure associated with the uniform distribution on $\lambda^d$, and if $B(x, r)$ is the closed ball centred at $x$ with radius $r$, then $\Lambda\left([B(x,r)]\right) = r$.*

*Proof:* By *Lemma 1*, the measure $\Lambda$ is translation invariant, and so $\Lambda([B(x,r)]) = \Lambda([B(0,r)])$. By the definition of product measures [1], $\varphi^{-1}[B(0,r)] = S^{d-1} \times [0, r]$, and so $\Lambda([B(0,r)])$ can be evaluated: $\Lambda([B(0,r)]) = \lambda^d(S^{d-1}) \cdot \lambda_+([0,r]) = r$.

# Appendix B: Supplementary figures and tables

Supplementary Figure 1: Boxplots for percent correct on all methods considered for the *SCZ42*, *SCZ51* and *SCZ93* datasets. Variance and disorganisation of these results may be due to noise, disorder and difficulty in the mapping from gene expression to schizophrenia. Despite this variance and disorganisation, many of the boxplots show significant improvements over the baseline. For example, a conservative sign test for the improvement of the wuRTF over the baseline in *SCZ93* is significant with a $p$-value of $5.1 \times 10^{-10}$.

| GL85 | BL | LR | SVM | RF | MRTF.i | uRTF.i | MRTF | uRTF | wMRTF | wuRTF |
|---|---|---|---|---|---|---|---|---|---|---|
| BL | | 6.7e-18 | 1.0e+00 | 7.0e-04 | 1.0e+00 | 1.0e+00 | 1.7e-24 | 1.0e+00 | 2.2e-28 | 5.3e-50 |
| LR | 6.7e-18 | | 6.7e-18 | 1.7e-24 | 2.3e-20 | 2.6e-17 | 2.6e-43 | 9.9e-20 | 5.3e-50 | 8.3e-55 |
| SVM | 1.0e+00 | 6.7e-18 | | 7.0e-04 | 1.0e+00 | 1.0e+00 | 1.7e-24 | 1.0e+00 | 2.2e-28 | 5.3e-50 |
| RF | 7.0e-04 | 1.7e-24 | 7.0e-04 | | 1.1e-01 | 4.6e-05 | 2.0e-07 | 2.0e-02 | 1.4e-14 | 1.2e-37 |
| MRTF.i | 1.0e+00 | 2.3e-20 | 1.0e+00 | 1.1e-01 | | 1.0e+00 | 2.3e-20 | 1.0e+00 | 1.4e-27 | 1.5e-44 |
| uRTF.i | 1.0e+00 | 2.6e-17 | 1.0e+00 | 4.6e-05 | 1.0e+00 | | 5.4e-26 | 1.0e+00 | 4.6e-30 | 5.3e-50 |
| MRTF | 1.7e-24 | 2.6e-43 | 1.7e-24 | 2.0e-07 | 2.3e-20 | 5.4e-26 | | 2.3e-22 | 7.0e-04 | 2.3e-20 |
| uRTF | 1.0e+00 | 9.9e-20 | 1.0e+00 | 2.0e-02 | 1.0e+00 | 1.0e+00 | 2.3e-22 | | 1.4e-27 | 1.5e-48 |
| wMRTF | 2.2e-28 | 5.3e-50 | 2.2e-28 | 1.4e-14 | 1.4e-27 | 4.6e-30 | 7.0e-04 | 1.4e-27 | | 3.2e-09 |
| wuRTF | 5.3e-50 | 8.3e-55 | 5.3e-50 | 1.2e-37 | 1.5e-44 | 5.3e-50 | 2.3e-20 | 1.5e-48 | 3.2e-09 | |

| SCZ42 | BL | LR | SVM | RF | MRTF.i | uRTF.i | MRTF | uRTF | wMRTF | wuRTF |
|---|---|---|---|---|---|---|---|---|---|---|
| BL | | 4.0e-08 | 1.0e+00 | 1.1e-01 | 9.9e-01 | 1.0e+00 | 8.2e-01 | 1.0e+00 | 1.8e-03 | 1.3e-05 |
| LR | 4.0e-08 | | 4.0e-08 | 3.8e-02 | 4.6e-05 | 4.2e-04 | 4.4e-03 | 6.9e-06 | 5.3e-01 | 3.1e-01 |
| SVM | 1.0e+00 | 4.0e-08 | | 1.1e-01 | 9.9e-01 | 1.0e+00 | 8.2e-01 | 1.0e+00 | 1.8e-03 | 1.3e-05 |
| RF | 1.1e-01 | 3.8e-02 | 1.1e-01 | | 5.8e-01 | 2.6e-01 | 8.9e-01 | 3.1e-01 | 9.6e-01 | 5.3e-01 |
| MRTF.i | 9.9e-01 | 4.6e-05 | 9.9e-01 | 5.8e-01 | | 1.0e+00 | 1.0e+00 | 9.6e-01 | 3.8e-02 | 5.2e-02 |
| uRTF.i | 1.0e+00 | 4.2e-04 | 1.0e+00 | 2.6e-01 | 1.0e+00 | | 6.4e-01 | 1.0e+00 | 6.9e-02 | 7.0e-04 |
| MRTF | 8.2e-01 | 4.4e-03 | 8.2e-01 | 8.9e-01 | 1.0e+00 | 6.4e-01 | | 7.4e-01 | 7.4e-01 | 1.8e-01 |
| uRTF | 1.0e+00 | 6.9e-06 | 1.0e+00 | 3.1e-01 | 9.6e-01 | 1.0e+00 | 7.4e-01 | | 6.9e-02 | 8.2e-05 |
| wMRTF | 1.8e-03 | 5.3e-01 | 1.8e-03 | 9.6e-01 | 3.8e-02 | 6.9e-02 | 7.4e-01 | 6.9e-02 | | 9.5e-01 |
| wuRTF | 1.3e-05 | 3.1e-01 | 1.3e-05 | 5.3e-01 | 5.2e-02 | 7.0e-04 | 1.8e-01 | 8.2e-05 | 9.5e-01 | |

| SCZ51 | BL | LR | SVM | RF | MRTF.i | uRTF.i | MRTF | uRTF | wMRTF | wuRTF |
|---|---|---|---|---|---|---|---|---|---|---|
| BL | | 1.1e-01 | 1.0e+00 | 7.6e-11 | 2.8e-03 | 1.0e+00 | 1.4e-13 | 1.0e+00 | 1.2e-15 | 4.7e-01 |
| LR | 1.1e-01 | | 1.1e-01 | 2.8e-02 | 9.3e-01 | 3.1e-01 | 1.1e-03 | 1.4e-04 | 8.2e-05 | 2.6e-01 |
| SVM | 1.0e+00 | 1.1e-01 | | 7.6e-11 | 2.8e-03 | 1.0e+00 | 1.4e-13 | 1.0e+00 | 4.1e-15 | 4.7e-01 |
| RF | 7.6e-11 | 2.8e-02 | 7.6e-11 | | 8.9e-02 | 4.0e-08 | 9.8e-01 | 1.4e-14 | 9.3e-01 | 6.9e-06 |
| MRTF.i | 2.8e-03 | 9.3e-01 | 2.8e-03 | 8.9e-02 | | 1.8e-01 | 2.8e-03 | 3.5e-06 | 8.2e-05 | 4.2e-01 |
| uRTF.i | 1.0e+00 | 3.1e-01 | 1.0e+00 | 4.0e-08 | 1.8e-01 | | 2.8e-11 | 8.2e-01 | 3.2e-09 | 9.6e-01 |
| MRTF | 1.4e-13 | 1.1e-03 | 1.4e-13 | 9.8e-01 | 2.8e-03 | 2.8e-11 | | 4.2e-19 | 1.0e+00 | 2.0e-07 |
| uRTF | 1.0e+00 | 1.4e-04 | 1.0e+00 | 1.4e-14 | 3.5e-06 | 8.2e-01 | 4.2e-19 | | 3.4e-16 | 2.8e-02 |
| wMRTF | 1.2e-15 | 8.2e-05 | 4.1e-15 | 9.3e-01 | 8.2e-05 | 3.2e-09 | 1.0e+00 | 3.4e-16 | | 4.4e-14 |
| wuRTF | 4.7e-01 | 2.6e-01 | 4.7e-01 | 6.9e-06 | 4.2e-01 | 9.6e-01 | 2.0e-07 | 2.8e-02 | 4.4e-14 | |

| SCZ93 | BL | LR | SVM | RF | MRTF.i | uRTF.i | MRTF | uRTF | wMRTF | wuRTF |
|---|---|---|---|---|---|---|---|---|---|---|
| BL | | 8.9e-02 | 1.0e+00 | 3.8e-02 | 9.6e-01 | 9.5e-01 | 6.9e-02 | 9.7e-01 | 6.6e-03 | 5.1e-10 |
| LR | 8.9e-02 | | 3.1e-01 | 7.8e-01 | 1.8e-01 | 9.7e-03 | 3.6e-01 | 8.9e-02 | 8.2e-01 | 3.6e-01 |
| SVM | 1.0e+00 | 3.1e-01 | | 3.6e-01 | 9.9e-01 | 1.0e+00 | 5.8e-01 | 1.0e+00 | 1.4e-01 | 4.0e-08 |
| RF | 3.8e-02 | 7.8e-01 | 3.6e-01 | | 3.1e-01 | 3.1e-01 | 9.1e-01 | 1.4e-01 | 9.3e-01 | 6.9e-02 |
| MRTF.i | 9.6e-01 | 1.8e-01 | 9.9e-01 | 3.1e-01 | | 9.1e-01 | 3.6e-01 | 7.8e-01 | 1.8e-03 | 2.0e-07 |
| uRTF.i | 9.5e-01 | 9.7e-03 | 1.0e+00 | 3.1e-01 | 9.1e-01 | | 2.6e-01 | 8.6e-01 | 3.8e-02 | 8.2e-05 |
| MRTF | 6.9e-02 | 3.6e-01 | 5.8e-01 | 9.1e-01 | 3.6e-01 | 2.6e-01 | | 2.2e-01 | 6.9e-01 | 2.8e-03 |
| uRTF | 9.7e-01 | 8.9e-02 | 1.0e+00 | 1.4e-01 | 7.8e-01 | 8.6e-01 | 2.2e-01 | | 1.4e-02 | 9.0e-08 |
| wMRTF | 6.6e-03 | 8.2e-01 | 1.4e-01 | 9.3e-01 | 1.8e-03 | 3.8e-02 | 6.9e-01 | 1.4e-02 | | 2.6e-01 |
| wuRTF | 5.1e-10 | 3.6e-01 | 4.0e-08 | 6.9e-02 | 2.0e-07 | 8.2e-05 | 2.8e-03 | 9.0e-08 | 2.6e-01 | |

Supplementary Table 1: Pairwise sign tests among all methods considered on *GL85*, *SCZ42*, *SCZ51* and *SCZ93* datasets. Each table shows raw sign test $p$-values indicated by methods in row and column headers (not corrected for multiple testing). Sign tests are conducted in a conservative manner, in which the sign test is one-tailed towards the better method, and ties are assigned in favour of the method that is not better. Bolding in Table 1 of the main text is found by examining the column of this table corresponding to the best method for a dataset, and then bolding every method in that column with $p$-value > 0.05 (*i.e.*, nominal significance). BL indicates a baseline method in which the mode label in the training dataset is predicted for all data items.

| Dataset | BL | LR | SVM | RF | MRTF.i | uRTF.i | MRTF | uRTF | wMRTF | wuRTF |
|---------|-----|------|------|------|--------|--------|-------|-------|-------|-------|
| *GL85*  | 5.58 | 10.52 | 5.58 | 7.18 | 6.09 | 5.16 | 6.43 | 5.78 | 6.16 | 6.05 |
| *SCZ42* | 10.62 | 13.89 | 10.60 | 12.17 | 10.93 | 11.10 | 11.76 | 11.40 | 11.70 | 11.41 |
| *SCZ51* | 10.65 | 12.77 | 10.67 | 11.74 | 11.36 | 11.12 | 9.84 | 10.21 | 11.07 | 9.69 |
| *SCZ93* | 7.74 | 9.65 | 7.24 | 7.42 | 7.20 | 6.94 | 7.65 | 7.24 | 6.86 | 6.90 |

Supplementary Table 2: Standard deviation of percent correct on gene expression datasets over 200 random train/test splits across different methods.

| Dataset | LR | SVM | RF | MRTF.i | uRTF.i | MRTF | uRTF | wMRTF | wuRTF |
|---------|------|------|-------|--------|--------|--------|--------|--------|--------|
| *GL85*  | <.08 | <.08 | 0.006 | 0.003 | 43.221 | 21.681 | 22.614 | 15.595 | 14.679 |
| *SCZ42* | <.08 | <.08 | 0.005 | 0.001 | 2.118 | 5.468 | 4.675 | 5.180 | 5.066 |
| *SCZ51* | <.08 | <.08 | 0.004 | 0.002 | 5.305 | 8.420 | 7.764 | 8.161 | 8.641 |
| *SCZ93* | 0.006 | 0.012 | 0.006 | 0.005 | 89.106 | 43.534 | 35.802 | 41.759 | 41.068 |

Supplementary Table 3: Comparison of mean running time (in minutes) on gene expression datasets over 200 random train/test splits across different methods. The experiments were run on an Intel Xeon CPU E5-2683v4@2.10GHz.

## Appendix C: Algorithm for Poisson thinning based exact inference

In this appendix, we provide a sampling strategy based on Poisson thinning within SMC that is more precise and efficient than the method displayed in Algorithm 2 of the main text. This strategy is listed in Supplementary Algorithm 1.

---

**Supplementary Algorithm 1** SMC for inferring RTP posteriors

---

1: **Inputs:** a) Training dataset $V$, $Z$, b) RTP measure $\Lambda$ on $H$, c) prespecified budget $\tau$, d) likelihood hyperparameter $\alpha$.
2: **Outputs:** Approximate RTP posterior $\sum_{m=1}^{M} \varpi_m \delta_{\triangle_{\tau,m}}$ at time $\tau$. ($\varpi_m$ are particle weights.)
3: Set $\tau_m \leftarrow 0$, for $m = 1, \ldots, M$.
4: Set $\triangle_{0,m} \leftarrow \{\text{hull } V\}$, $\varpi_m \leftarrow 1/M$, for $m = 1, \ldots, M$.
5: **while** $\min\{\tau_m\}_{m=1}^{M} < \tau$ **do**
6:     Resample $\triangle'_{\tau_m,m}$ from $\{\triangle_{\tau_m,m}\}_{m=1}^{M}$ *w.p.p.t.* $\{\varpi_m\}_{m=1}^{M}$, for $m = 1, \ldots, M$.
7:     Set $\triangle_{\tau_m,m} \leftarrow \triangle'_{\tau_m,m}$, for $m = 1, \ldots, M$.
8:     Set $\varpi_m \leftarrow 1/M$, for $m = 1, \ldots, M$.
9:     **for** $m \in \{m : m = 1, \ldots, M \text{ and } \tau_m < \tau\}$ **do**
10:         Set $\tau' \leftarrow \infty$, $h \leftarrow \varnothing$, $a \leftarrow \varnothing$.
11:         Set $\tau_{a'} \leftarrow \tau_m \ \forall \ a' \in \triangle_{\tau_m,m}$.
12:         **while** 1 **do**
13:             Set $\tau^* \leftarrow \infty$.
14:             **for** $a' \in \triangle_{\tau_m,m}$ **do**
15:                 **if** $\tau_{a'} < \tau'$ **then**
16:                     Sample $\delta \sim \text{Exp}(r_{a'})$.
17:                     Sample $h' \sim \Lambda(\cdot \cap [B(r_{a'})])/\Lambda([B(r_{a'})])$.
18:                     Set $\tau_{a'} \leftarrow \tau_{a'} + \delta$.
19:                     **if** $\tau_{a'} < \tau'$ **and** $h' \cap a' \neq \varnothing$ **then**
20:                         Set $\tau' \leftarrow \tau_{a'}$, $h \leftarrow h'$, $a \leftarrow a'$.
21:                     **if** $\tau_{a'} < \tau^*$ **and** $h' \cap a' = \varnothing$ **then**
22:                         Set $\tau^* \leftarrow \tau_{a'}$.
23:             **if** $\tau^* \geq \min\{\tau', \tau\}$ **then**
24:                 **break**
25:         Set $\triangle_{t,m} \leftarrow \triangle_{\tau_m,m}$, for all $t \in (\tau_m, \min\{\tau, \tau_m + \tau'\}]$.
26:         **if** $\tau_m + \tau' \leq \tau$ **then**
27:             Set $\triangle_{\tau_m,m} \leftarrow (\triangle_{\tau_m,m}/\{a\}) \cup \{\text{hull}(V \cap a \cap h^-), \text{hull}(V \cap a \cap h^+)\}$.
28:             Set $\varpi_m \leftarrow \varpi_m P(Z|\triangle_{\tau_m,m}, V, \alpha)/P(Z|\triangle'_{\tau_m,m}, V, \alpha)$.
29:         **else**
30:             Set $\triangle_{t,m} \leftarrow \triangle_{\tau_m,m}$, for $t \in (\tau_m, \tau]$.
31:         Set $\tau_m \leftarrow \tau_m + \tau'$.
32:     Set $\mathcal{Z} \leftarrow \sum_{m=1}^{M} \varpi_m$.
33:     Set $\varpi_m \leftarrow \varpi_m/\mathcal{Z}$, for $m = 1, \ldots, M$.
34: **return** the particle approximation $\sum_{m=1}^{M} \varpi_m \delta_{\triangle_{\tau,m}}$.

---

Here, on lines 10 to 24, we provide Poisson thinning for the cutting events. Also, line 12 is an infinite loop, broken by lines 23 and 24 (*i.e.*, it is a **do/until** loop). We note that the computational cost of this method may be reduced by sorting the $a'$ in the loop on line 14 according to the expectation (or a realized value) of the $\tau_{a'}$, or by organising the loops on lines 12 and 14 in a different way in order to respect the cut probabilities. On line 27, $h^-$ and $h^+$ refer to the two closed half-spaces bounded by $h$.

Differing from Algorithm 2 in the main text, instead of sampling the time to the next cut, and then choosing the involved polytope *w.p.p.t.* the cut intensities, we record cuts for all polytopes and find the earliest cut (after thinning). We then discard all future cuts before continuing (making use of the 'add-min' nature of the exponential distribution, as is standard in MJP work). In the thinning code, the variable $\tau'$ records the time to the earliest cut, and $\tau^*$ records the minimum of the budgets expended by each of the polytopes that have not yet been cut. For more detail on Poisson thinning, we refer to Chapter 2 of [2].

**Appendix D: Manual for `tess19` v1.0**

**Copyright (c) 2019. Shufei Ge and Lloyd T. Elliott**

The `tess19` software implements the Bayesian nonparametric methods described in Ge et al., *Random Tessellation Forests*, 2019. This software constructs a random forest for posterior prediction of categorical data based on real valued predictors. The trees of the random forest are found through SMC inference. This manual is for version v1.0. This software requires the following R packages: `optparse`, `purrr`. This software is released under the open source BSD 2-clause license.

**1. Basic Usage**

```
tess19 <IFILE.txt> <OFILE.txt>
```

Predictions for the missing labels in the `Levels` column of the file `<IFILE.txt>` are made using the predictors in the file `<IFILE.txt>` and the uRTF model, with 100 trees, with the prespecified budget $\tau = \infty$, and with the hyperparameter settings $\alpha_k = n_k/1000$. The predictions are saved in the `Levels` column of the file `<OFILE.txt>`.

The format of the file `<IFILE.txt>` is as follows. The file is space separated. The first line is a header line with one column name for each of predictor (for example, `V1 V2 ...`), followed by a column named `Levels`. Subsequent lines are given with one line per data item, with the predictors for the data item followed by the items level. The predictors are real numbers and the levels must be positive integers in the set $1, \ldots K$, where $K$ is the number of levels. Both test and train data must be provided in `<IFILE.txt>`, and the test data items must have missing labels indicated by the string `NaN`. The predictors may not have missing data.

The format of the file `<OFILE.txt>` is as follows. The first line is a header line naming the column `Levels` (*i.e.*, one column). Subsequent lines are given with one line for data item. If the data item is a training data item, then the value `NaN` is recorded in the corresponding line. If the data item is a testing data item, then a predicted label is recorded.

**2. Advanced Usage**

```
tess19 --usage

tess19 --license

tess19 --version
```

```
tess19 [--Mondrian] [--weights <WFILE.txt>] [--cuts <MAX-CUTS>]
[--tau <PRESPECIFIED-BUDGET>] [--alpha <HYPER-PARAMETER>] [--trees
<NUMBER-OF-TREES>] [--particles <PARTICLES>] [--seed <SEED>]
<IFILE.txt> <OFILE.txt>
```

`--usage`. Prints this manual to the standard output.

`--license`. Prints the open source BSD 2-clause license for this software.

`--version`. Prints the software version information.

`--Mondrian`. Instructs `tess19` to conduct axis aligned cuts, yielding the MRTF model, or (if the `--weights` flag is provided) the wMRTF model. By default, axis aligned cuts are not used.

`--weights <WFILE.txt>`. Instructs `tess19` to use a weighted version of the uniform distribution for the measure $\lambda^d$, yielding the wuRTF or wMRTF model. The weights $\omega_j$ are read from the file `<WFILE.txt>` which must contain $d$ lines corresponding to the prior weight for each predictor, with one real number per line.

`--cuts <MAX-CUTS>`. This flag sets a stopping condition wherein SMC particles will return after `<MAX-CUTS>` cuts, regardless of the budget. By default, `<MAX-CUTS>` is set to 100. This value must be a positive integer. By the pausing condition, each cut separates a data item, and setting `<MAX-CUTS>` to a value larger than or equal to the number of data items is equivalent to setting `<MAX-CUTS>` to $\infty$.

`--tau <PRESPECIFIED-BUDGET>`. This flag specifies the budget. The budget $\tau$ must be a positive real number, or $\infty$ (the string `Inf`). By default, $\tau$ is set to infinity.

`--alpha <HYPER-PARAMETER>`. This flag is a positive real number that sets the coefficient of the empirical label proportion in the Dirichlet/multinomial prior, so that the value of the hyperparameter is $\alpha_k = $ `<HYPER-PARAMETER>` $* \; n_k$. The default value is $10^{-3}$.

`--ntrees <NUMBER-OF-TREES>`. This flag is a positive integer that sets the number of trees to use in the random forest. By default, `<NUMBER-OF-TREES>` is set to 100. A value of 1 specifies the uRTP/wuRTP/MRTP/wMRTP priors (i.e., no random forest).

`--particles <PARTICLES>`. This flag sets the number of particles to use in the SMC approximations. The default value of `<PARTICLES>` is 100.

`--seed <SEED>`. This flag sets the random seed to `<SEED>`. The default is to use the system clock to set the random seed.

### 3. License

```
tess19 v1.0.  Copyright (c) 2019.  Shufei Ge and Lloyd T. Elliott.

Redistribution and use in source and binary forms, with or without
modification, are permitted provided that the following conditions are met:

1.  Redistributions of source code must retain the above copyright notice,
this list of conditions and the following disclaimer.

2.  Redistributions in binary form must reproduce the above copyright
notice, this list of conditions and the following disclaimer in the
documentation and/or other materials provided with the distribution.
THIS SOFTWARE IS PROVIDED BY THE COPYRIGHT HOLDERS AND CONTRIBUTORS "AS IS"
AND ANY EXPRESS OR IMPLIED WARRANTIES, INCLUDING, BUT NOT LIMITED TO, THE
```