[Reviews · NeurIPS 2019]

Reviewer 1



The paper is a novel generalization of the well-known Mondrian Process. However, some of the portions could have been explained in words to make the paper more readable. The content appears original and it provides a convenient and general approach to classification. Although not much is discussed about the large sample consistency of the method, the approach appears to be intuitive in itself.

Reviewer 2



The authors present a very interesting new extension to random partition type priors with an appealing inference scheme. I think it is a solid extension of existing work. Numerical results suggest that it can perform just as well or better than existing random partition models, though I'm curious about the comparison of runtimes against various methods. One other thing I noticed is that you choose the particle with the highest weight after the SMC procedure. It seems like prediction would be improved by using an average of the prediction from each particle with respect to the particle weight. Is there a more principled reason as to why you do not do this? I think a discussion of your method with in comparison BART type methods would be helpful, as many people are more familiar with BART than the Mondrian method, and could help the reader unfamiliar with this literature. After reading the author rebuttals and the other reviews comments, I am inclined to vote "accept" for this paper. I think the idea is quite novel and well written. I think the addition of comparisons, even in the supplementary materials will help strengthen this paper. Also, please add standard errors for the results in Table 1.

Reviewer 3



General metrics: Originality: Yes this is an original contribution. The authors have also properly highlighted the context in which it sits. Quality: This work is high quality and polished and well presented. Some improvements could be made, see comments below. Clarity: The work is clearly presented, the figures and algorithms are useful. A handful of minor presentation issues highlighted below. Significant: The work feels like a significant new contribution and I can envision practitioners using these ideas and other theoretical research building on this work. Major comments 1. The idea presented is theoretically interesting and novel in an ML context. I thank the authors for a pleasurable read. 2. The approach would only seem to work in a bounded domain. (Since the measure $\Lambda$ is partly made up of Lebesgue measure on the half-line, $\Lambda([a])$ the normalising constant would be infinite otherwise.) This is stated by the authors at the head of the algorithm though possibly could be highlighted more clearly. eg. on line 61 it should read $W \subset R^d$ (ie strict subset). 3. I was sceptical that rejection sampler would work as written in a space of even moderately high dimension. As the dimension grows, does the smallest ball containing the convex hull of the data not increase in relative volume to an infeasible level? Or does the hyperplane (of dim d-1) still intersect a with reasonable probability? The example in Sec 3.2 seems to suggest it's still fine. Some reference to this problem, and the performance of the rejection sampler in relation to the dimension of the problem, should be included. 4. Similarly the authors state eg on line 168/169 that explicit computation of the polytopes a is not required. In this way they do not have access to the the value of $\Lambda([a])$, which is the normalising constant required in Algorithm 1. They state instead that they approximate the value with the volume of the covering ball. Is this approximation not increasingly poor in higher dimensions? Please refer to this issue, possibly in conjunction with the previous point. 5. Line 170-173. This seems very strange. What if lots of the test data is outside the collection of convex polytopes? Isn't the whole point of the inference to label the test data rather than average over the possible values of the labels? Or is this just used to form possibly larger convex hulls to include the test data? (In which case, this does not seem very robust in the case of inference for a second test dataset.) I am possibly misunderstanding what you are doing here but either way this section requires significant clarification. 6. line 206. The tesselations in T will not be equal wp1. So what is the meaning of the mode of T? Is it simply the single tesselation with highest marginal likelihood? Please explain more clearly in the text what you are choosing. 7. What is the effect of setting the Dirichlet parameter $\alpha$ as you have in line 219. Are there higher level features of the model that would change based on the setting of this value? cf Polya trees where eg continuous or discrete distributions can result depending on simply the value chosen for the equivalent parameter. Minor & notational comments 1. line 61 define Y. (I think you should just put 'Y' between the words colctionandof'). 2. line 86 H in italics 3. line 93 $\lambda^d$ should be superscript 4. line 133 P(Z) does not appear in the preceding equation. Do you mean P(V,Z)? Or P(V|Z)P(Z)? 5. line 139 what is $\phi^{-1}?$ Define. 6. line 146 Is it not sufficient to sample $u$ from [0,r]? As written this contradicts the definition of u earlier. 7. line 161. The notation of intersection with V is confusing (refers to intersection with convex hull). Your own notation on line 16 of algorithm 2 is much clearer, suggest you use that. 8. line 163 insert "strict" between "a" and "subset". 9. line 219. typo "to \alpha to n" 10. line 229. what does centre the points mean? write mathematically? 11. line 232. which category is coloured red and which black? (would be easier if you just stated it) 12. line 259-264 several typos principle -> principal. also analysis. 13. line 268 what is uRTF.i and MRTF.i? not defined anywhere. 14. line 272 if the budget is infinite, when does the partitioning procedure stop? UPDATE AFER RESPONSE: Thanks for detailed response. Please include the updated polytope sampling procedure (comment 3) in the final version. While the dimensionality issue still exists I am satisfied the authors are not sweeping it under the carpet and it is good that they are attempting to find strategies to deal with it. Explaining these clearly in the final submission would improve it. Include the D=85 figure to answer any similar questions raised by future readers. I am still not totally clear about the response to Comment 5. What is the effect of `snapping to the nearest'? This seems like it could be a significant source of error. Please further explain the assertion wrt missing-at-random data in the final submission. Please also include in the final submission the clarifications asked for in the minor comments which you did not specifically address in your response (presumably due to space constraints).

[Author Response · NeurIPS 2019]

We thank the reviewers for their comments and suggestions. Many of the comments are quite good and will improve the quality of the paper. Minor comments and typos have now been fixed in the text, and we thank the reviewers for pointing them out. Our point-by-point response to the reviewers' major comments follows, with their comments italicized.

**Reviewer 1:** ... *not much is discussed about the large sample consistency of the method.* As far as we know, the identifiability of the Mondrian process in the infinite data limit is still an open problem. If such a theory were discovered, it would likely generalise to random tessellation processes. There is however work showing that Mondrian forests achieve minimax convergence rates for regression (Mourtada et al. 2018), and in future work those proofs may be adapted to random tessellation processes. We have added this discussion to the manuscript.

**Reviewer 2:** ... *I'm curious about the comparison of runtimes against various methods.* We display below the mean running time (in minutes) across different methods for the largest dataset *SCZ93* (left table) and runtimes for the wuRTF on all datasets (right table). The experiments were run on an Intel Xeon CPU E5-2683v4@2.10GHz.

| Dataset | LR | SVM | RF | MRTF.i | uRTF.i | MRTF | uRTF | wMRTF | wuRTF |
|---|---|---|---|---|---|---|---|---|---|
| *SCZ93* | 0.006 | 0.012 | 0.006 | 0.005 | 89.106 | 43.534 | 35.802 | 41.759 | 41.068 |

| Dataset | *SCZ42* | *SCZ51* | *GL85* | *SCZ93* |
|---|---|---|---|---|
| Runtime | 5.066 | 8.641 | 14.679 | 41.068 |

While the runtimes are not the focus of the paper, we will include a complete version of these tables in the supplement. Further, since submitting this work, we've developed new inference based on pseudomarginals allowing the spherical approximation and rejection sampling to be replaced by a scheme allowing exact samples without computation of $\lambda^{d-1}([a])$. Finding the radius of the sphere and rejection sampling are the bottlenecks for our methods, and so this advance will considerably improve the runtimes of RTF methods.

**Reviewer 3**

*Comment 2: The approach would only seem to work in a bounded domain...* $W$ is compact, and so we should have used a strict subset. This is now corrected in the text. We note that projective processes can be extended to unbounded domains using Kolmogorov extension theorems. This is done for example in the Nagel and Weiss reference, but we do not consider it as it is not relevant for inference.

*Comment 3: I was skeptical that a rejection sampler would work as written in a space of even moderately high dimension ... does the hyperplane ... still intersect a with reasonable probability? & Comment 4: Similarly the authors state eg on line 168/169 that explicit computation of the polytopes a is not required ... Is this approximation not increasingly poor in higher dimensions?* Rejection sampling and computing the radius of the approximating sphere are computational bottlenecks (due to the reasons raised by the reviewer). This leads to longer runtimes. For $D = 85$, we are able to conduct inference with the rejection sampling, indicating that the interaction is still possible at this dimensionality. Since submitting this work, we've improved inference using a pseudomarginal method in which, instead of choosing a polytope to cut with probability proportional to the radius of a sphere, we instead sample a hyperplane cutting the whole domain and choose a polytope to cut uniformly among all polytopes that the cut intersects. This obviates the need for approximations and rejection sampling. We will discuss or report on this new method in the camera ready copy, should this work be accepted.

*Comment 5: What if lots of the test data is outside the collection of convex polytopes?* When we form convex hulls, we consider a version of the training data that includes the predictors of the testing data. Testing data lying outside of the convex hulls formed by the training data will be 'snapped to the nearest' polytope. Further, when data are missing-at-random, test data will not generally lie outside of the convex polytopes formed by the training data and so not much data will be snapped in this way. Data affected by test/train shifts may suffer from this approach. However, test/train shifts tend to confound any machine learning method and so this analysis is outside the scope of the paper.

*Comment 6: The tessellations in T will not be equal wp1. ... what is the mode of T?* For each random tessellation process in a forest, we predict a label for each test point. By 'mode', we meant to refer to the mode of the distribution of the predicted label, and not the of the tessellations themselves. We've now clarified this in the text.

*Comment 7: What is the effect of setting the Dirichlet parameter $\alpha$ as you have in line 219...* Our setting of $\alpha$ provides a weak prior matched to the empirical label frequencies. We do not use any higher level features. This method is popular for likelihoods in Bayesian nonparametrics. An exploration of this likelihood and also hierarchical likelihoods and Polya trees are an area of future work.

*Minor comment 6: Is it not sufficient to sample $u$ from [0,r]?* This was a typo. It's indeed sufficient and correct to sample $u$ from $[0, r]$, and we do that in our implementation.

*Minor comment 14: If the budget is infinite, when does the partitioning procedure stop?* We use a 'pausing condition' that is described in Section 2.2.2 and originally proposed in Breiman 2001: if the training points in a polytope all have the same label then no further cuts are performed on the polytope.

[Meta-Review · NeurIPS 2019]

All reviews for this paper are positive and the authors response looks convincing. I think the paper should therefore be accepted. Personally, while I like very much the proposed approach and I agree that the technical contribution is strong, I'm a bit disappointed by the empirical validation: while some improvements are shown with respect to existing baselines, only very small, non standard, datasets are considered and methods are only compared in terms of classification error, which does not really allow to show the benefit of using a Bayesian framework. I'm also missing among the baselines non bayesian forests of oblique trees, since the possibility to consider non axis-parallel splits seems to be one of main benefits of the approach. Any attempt to improve the empirical validation along these lines would be greatly appreciated and would clearly strengthen the paper I think.